# Iron-Utilization System in *Vibrio vulnificus* M2799

**DOI:** 10.3390/md19120710

**Published:** 2021-12-17

**Authors:** Katsushiro Miyamoto, Hiroaki Kawano, Naoko Okai, Takeshi Hiromoto, Nao Miyano, Koji Tomoo, Takahiro Tsuchiya, Jun Komano, Tomotaka Tanabe, Tatsuya Funahashi, Hiroshi Tsujibo

**Affiliations:** 1Department of Microbiology and Infection Control, Faculty of Pharmacy, Osaka Medical and Pharmaceutical University, 4-20-1 Nasahara, Takatsuki, Osaka 569-1094, Japan; kawano@gly.oups.ac.jp (H.K.); okai@gly.oups.ac.jp (N.O.); hiromoto@gly.oups.ac.jp (T.H.); miyano@gly.oups.ac.jp (N.M.); takahiro.tsuchiya@ompu.ac.jp (T.T.); jun.komano@ompu.ac.jp (J.K.); hiroshi.tsujibo@ompu.ac.jp (H.T.); 2Department of Physical Chemistry, Faculty of Pharmacy, Osaka Medical and Pharmaceutical University, 4-20-1 Nasahara, Takatsuki, Osaka 569-1094, Japan; koji.tomoo@ompu.ac.jp; 3Laboratory of Hygienic Chemistry, College of Pharmaceutical Sciences, Matsuyama University, 4-2 Bunkyo-cho, Matsuyama, Ehime 790-8578, Japan; ttanabe@g.matsuyama-u.ac.jp (T.T.); tfunahas@g.matsuyama-u.ac.jp (T.F.)

**Keywords:** siderophore, periplasmic binding protein, siderophore-interacting protein, ferric-siderophore reductase, aerobactin, desferrioxamine B

## Abstract

*Vibrio vulnificus* is a Gram-negative pathogenic bacterium that causes serious infections in humans and requires iron for growth. A clinical isolate, *V*. *vulnificus* M2799, secretes a catecholate siderophore, vulnibactin, that captures ferric ions from the environment. In the ferric-utilization system in *V*. *vulnificus* M2799, an isochorismate synthase (ICS) and an outer membrane receptor, VuuA, are required under low-iron conditions, but alternative proteins FatB and VuuB can function as a periplasmic-binding protein and a ferric-chelate reductase, respectively. The vulnibactin-export system is assembled from TolCV1 and several RND proteins, including VV1_1681. In heme acquisition, HupA and HvtA serve as specific outer membrane receptors and HupB is a sole periplasmic-binding protein, unlike FatB in the ferric-vulnibactin utilization system. We propose that ferric-siderophore periplasmic-binding proteins and ferric-chelate reductases are potential targets for drug discovery in infectious diseases.

## 1. Introduction

Iron is essential for all living organisms and human-pathogenic bacteria require ferrous ions for their growth and virulence [1]. However, the biologically available amount of iron is restricted because in the oxidative atmosphere at physiological pH, ferrous ions (Fe^2+^) are quickly oxidized to ferric ions (Fe^3+^). In aqueous solution, iron is present as dissolved Fe^3+^ (10^−10^ M) at physiological pH [2]. However, in the human body, most Fe^3+^ is tightly bound to iron-binding proteins, such as transferrin, lactoferrin and ferritin [3], and the level of freely available iron (10^−18^ M) is insufficient to sustain bacterial life. Therefore, human pathogenic bacteria produce highly specific iron-utilization systems, which include siderophores and specific receptors. Most bacteria secrete siderophores, which are low-molecular weight metal-chelators that capture Fe^3+^ outside the cell [4]. In general, these siderophores are categorized as catecholate or hydroxamate types [5]. The iron-utilization systems are candidates for new drug discovery in bacterial infectious diseases.

*Vibrio vulnificus* is a Gram-negative halophilic marine bacterium that is highly dependent on iron for growth. *V*. *vulnificus* causes septicemia and serious wound infections and symptoms such as fever, nausea, abdominal pain and hypertension [6]. Primary septicemia is often associated with patients who have increased iron concentrations from chronic liver damage [7]. The virulence factors of *V*. *vulnificus* include a metalloproteinase [8], a hemolysin [9], a capsular polysaccharide [10], siderophore dependent iron-acquisition systems [11], a heme-acquisition system [12] and RTX toxin [13]. Recently, IscR has been reported to positively regulate *vvhBA*, encoding a cytolysin/hemolysin of *V*. *vulnificus*, by sensing nitrosative stress and iron starvation [14]. Among these factors, the iron-acquisition systems play critical roles in *V*. *vulnificus* infections [11,15]. Roig et al. had divided *V*. *vulnificus* into five phylogenetic lineages (L) according the analysis of the core genome [16]. In clinical isolates, *V*. *vulnificus* CMCP6, MO6-24 and YJ016 are classified as L1, while ATCC27562^T^ is a member of L2.

We previously reported that *V*. *vulnificus* M2799, a clinical isolate, has 100-fold higher lethality in mice compared to that of an environmental isolate, strain JCM3731 [17]. *V*. *vulnificus* M2799 showed much greater cytotoxicity than strain JCM3731 towards various cultured cells [17]. In mice inoculated with strain M2799 or JCM3731, the number of neutrophils increased, whereas strain M2799 reduced the number of macrophages and strain JCM3731 had no effect. However, the pathogenesis of strain M2799 is not completely elucidated. *V. vulnificus* M2799 secretes vulnibactin to capture Fe^3+^ from the environment [18]. Vulnibactin is a catecholate siderophore that is assembled from one molecule of 2,3-dihydroxybenzoic acid, two molecules of salicylic acid and two molecules of L-threonine. Vulnibactin is synthesized via isochorismate and is dependent on non-ribosomal peptide synthases [19]. Isochorismate synthase (ICS) and isochorismatase (VenB) play a key role in vulnibactin biosynthesis [11] and the vulnibactin-mediated iron-utilization system is critical for use of transferrin-bound iron and the virulence of *V*. *vulnificus* [11,20].

Vulnibactin is biosynthesized in the cytoplasm of *V*. *vulnificus* and translocates to extracellular spaces, including in infection of human tissues (Figure 1). Vulnibactin chelates Fe^3+^ in the environment and the ferric-vulnibactin is imported to the periplasm through a specific outer membrane receptor, VuuA (Figure 1) [21]. The ferric-vulnibactin in the periplasm is captured by FatB, a periplasmic-binding protein (PBP) and transported through the inner membrane by an ATP binding cassette (ABC) transporter [22]. In the cytoplasm, ferric-vulnibactin is reduced by VuuB, a ferric-chelate reductase (FCR) that is a member of the flavin adenine dinucleotide (FAD)-containing siderophore-interacting protein (SIP) family (Figure 1) [22].

In this study, we focus on the iron-utilization system in *V*. *vulnificus* M2799 and we propose new targets for antimicrobial agents in this system.

## 2. Proteomic Analysis under Iron-Repleted and Low-Iron Conditions

First, we used a proteomic approach to study differential expression of proteins from *V*. *vulnificus* M2799 under iron-repleted and low-iron conditions, using two-dimensional differential gel electrophoresis (2D-DIGE) [23]. To investigate growth phase-dependent protein expression, 2D-DIGE analyses were carried out using whole-cell lysates grown until the early, mid and late logarithmic growth phases. About 2500 spots were detected on each gel image. After gel-to-gel matching, spots that showed significant differences of ≥2.0-fold (*p* ≤ 0.01) in expression level between iron-repleted and low-iron conditions were subjected to in-gel digestions with trypsin and then characterized by peptide mass fingerprinting (PMF). Proteins were identified using matrix-assisted laser desorption/ionization-time of flight mass spectrometry (MALDI-TOF-MS).

Iron-regulated 32, 53 and 42 spots in the lysates were found in the early, mid and late logarithmic growth phases, respectively, from which 18, 31 and 26 different proteins were identified by MS [23]. The Mascot search engine (Matrix Science) was used for PMF analysis. At that point, only two genomes from *V*. *vulnificus* CMCP6 and YJ016 had been clarified. PMF analyses indicated that *V*. *vulnificus* M2799 was a close relative of *V*. *vulnificus* CMCP6. The predicted iron-utilization systems in *V*. *vulnificus* M2799 are shown schematically in Figure 1. Yellow-boxed letters show the proteins identified in PMF analyses. Many of these proteins were involved in the ferric-vulnibactin utilization system (Figure 1). To acquire iron from the environment, *V*. *vulnificus* M2799 utilizes ferric-vulnibactin, ferric-aerobactin, ferrioxamine B, heme and free ferrous and ferric ions. The structures of specific iron chelators utilized by *V*. *vulnificus* M2799 are shown in Figure 2. Aerobactin and desferrioxamine B are hydroxamate-type siderophores that are synthesized by other organisms (Figure 2). The ferric-aerobactin and ferrioxamine B utilization systems in *V*. *vulnificus* M2799 have also been identified in previous studies [24,25].

## 3. Growth of the Deletion Mutants under Low-Iron Conditions

Deletion mutants were constructed for genes encoding proteins in the ferric-vulnibactin utilization system, including ICS, VuuA, FatB and VuuB (Figure 1). Heart infusion broth containing 2% NaCl with the iron chelator ethylenediamine-di(*o*-hydroxyphenylacetic acid) (EDDA) at a final concentration of 10 μg/mL was used for growth examination under low-iron conditions. Unlike the isogenic wild-type strain, the ICS and VuuA mutants (Δ*ics* and Δ*vuuA*) were unable to grow under low-iron conditions [22]. This confirmed that the ferric-vulnibactin utilization system has an important role in growth of *V*. *vulnificus* M2799 in low-iron conditions. Moreover, ICS is an essential enzyme in vulnibactin biosynthesis and VuuA is the sole receptor in the ferric-vulnibactin import system. In contrast, FatB and VuuB mutants showed retarded growth under low-iron conditions, indicating that there must be alternative proteins in *V*. *vulnificus* M2799 that can complement the loss of function of FatB and VuuB [22]. Fur, a transcriptional repressor that responds to iron concentrations, represses expression of genes involved in iron-utilization systems under iron-replete conditions [26,27]. In the Δ*fur* mutant, the expression levels of *ics*, *vuuB*, *fatB*, *vuuA* and *vatD* under both iron-replete and low-iron conditions were similar to those in the wild-type strain under low-iron conditions. These results show that these genes were negatively regulated by Fur [22].

## 4. VatD in the Ferric-Vulnibactin Utilization System

A search for the alternative protein to FatB in the ferric-vulnibactin utilization system in *V*. *vulnificus* M2799 using BLAST revealed that FatB is homologous to VatD (VV2_1012), a putative ferric-aerobactin and ferrioxamine B PBP, in the *V*. *vulnificus* CMCP6 genome. VatD showed sequence similarity (17% identity) with FatB. To examine whether VatD is used instead of FatB in strain M2799, a double mutant of *fatB* and *vatD* (Δ*fatB*Δ*vatD*) was constructed. This double mutant showed further growth impairment compared with the single mutant (Δ*fatB*) under low-iron conditions, whereas Δ*vatD* showed the same growth curve as the wild-type [22]. To confirm these findings, *vatD* was cloned into pRK415 [28] and complementation analysis was performed. Growth of Δ*fatB*Δ*vatD* complemented by *vatD* was restored to the level of Δ*fatB*. These results show that VatD, which functions as a ferric-hydroxamate siderophore PBP, participates in the ferric-vulnibactin utilization system in the absence of FatB (Figure 3a) [22].

To analyze the structures of FatB and VatD, an expression system was constructed for both proteins. FatB and VatD were expressed as His-tag fusion proteins and purified to high purity using a Ni-chelate affinity and gel filtration chromatographies. Crystallization conditions for FatB and VatD were examined using several commercial screens and good crystals of apo VatD and the VatD-ferrioxamine B complex were obtained [29]. However, good crystal of FatB was not obtained. Structure of the VatD-ferrioxamine B complex was solved by the molecular replacement method using the MOLREP program (PDB ID: 7W8F). The electron density of ferrioxamine B bound to VatD was clearly apparent in the early stage of structural refinement. The structure of VatD has two domains (N- and C-domains) connected by two rigid long helices. The ferrioxamine B binding site is formed by the N- and C-domains of VatD. We also have determined the structure of the apo-VatD at 2.6 Å resolution and the structural analysis at a higher resolution is in progress. Comparison of the VatD-ferrioxamine B complex with the apo form revealed significant movement of the C-domain in the complex upon binding of ferrioxamine B (Figure 4).

## 5. IutB in the Ferric-Vulnibactin Utilization System

VuuB is a ferric-vulnibactin FCR in the FAD-containing SIP family that has been shown to be complemented by other reductases in bacteria with defective *vuuB* [30]. To find genes encoding an alternative ferric-vulnibactin FCR to VuuB, we searched for genes encoding putative oxidoreductases located in gene clusters for iron-utilization systems. This search identified *iutB* (VV2_1010) and *desB* (VV2_1339) in gene clusters for the ferric-aerobactin (*vatCDB*) and ferrioxamine B (*desRA*) utilization systems, respectively [30]. The *iutB* gene is located upstream of *vatCDB*, whereas *desB* is located upstream of *desRA* in the opposite direction. In the gene clusters, IutR and DesR are coded as GntR-type negative and AraC-type positive regulators, respectively (Figure 1) [24,25,31]. IutB and DesB are ferric-hydroxamate siderophore reductases (FSRs), similar to FhuF in *Escherichia coli*.

Deletion mutants of *vuuB* and *iutB* were then constructed. The Δ*vuuB*Δ*iutB* double mutant exhibited further growth impairment compared with the Δ*vuuB* single mutant under low-iron conditions, while Δ*iutB* showed the same growth curve as the wild-type [30]. Furthermore, to verify directly that VuuB and IutB reduce ferric-vulnibactin, we confirmed the growth of deletion mutants in the Δ*ics* strain under low-iron conditions in the absence or presence of vulnibactin. This strain cannot synthesize vulnibactin and then is not able to grow under low-iron conditions, while addition of vulnibactin recovers growth. Although growth of Δ*ics*Δ*vuuB* was slower than that of Δ*ics* in the presence of vulnibactin, Δ*ics*Δ*iutB* showed the same growth curve as that of the Δ*ics* strain. The triple mutant Δ*ics*Δ*vuuB*Δ*iutB* exhibited further growth impairment compared with the Δ*ics*Δ*vuuB* strain [32]. Therefore, VuuB was shown to be a major ferric-vulnibactin reductase and IutB functions in the absence of VuuB (Figure 3a). However, DesB was not involved in the ferric-vulnibactin utilization system in M2799 [30]. VuuB and DesB can play a role in ferric-aerobactin reduction in the absence of IutB (Figure 3a) [30].

To examine the molecular mechanisms of ferric reduction by VuuB and IutB, expression systems were constructed for purification of these proteins. FCRs catalyze reduction of Fe^3+^ to release Fe^2+^, using nicotinamide adenine dinucleotide (NADH), nicotinamide adenine dinucleotide phosphate (NADPH), or glutathione (GSH) as an electron donor. In some bacterial FCRs, reduction of oxidized FAD (FAD_ox_) to reduced FAD (FAD_red_) was coupled with reduction by NAD(P)H. For FCR activity, VuuB prefers NADH to NADPH and this activity was enhanced by FAD, indicating that FAD functions as a cofactor. In the presence of 10 μM FAD and 20 μM NADH, VuuB showed specific FCR activity with Fe^3+^-NTA as an electron acceptor, whereas IutB exhibited specific FCR activity in the presence of 100 μM GSH and 100 μM NADPH [32].

The FCR activities of VuuB and IutB were investigated using ferric-hydroxamate siderophores (ferric-aerobactin and ferrioxamine B) and ferric-catecholate siderophores (ferric-enterobactin, ferric-vibriobactin and ferric-vulnibactin). VuuB and IutB were found to catalyze reduction of ferric-aerobactin, ferric-vibriobactin and ferric-vulnibactin, showing that VuuB and IutB reduce both ferric-catecholate and ferric-hydroxamate siderophores. With use of ferric-vulnibactin as an electron acceptor, the specific FCR activity of VuuB was a little lower than that of IutB [32]. Furthermore, expression of *vuuB*, *iutB* and *desB* increased in the wild-type strain under low-iron conditions, with the expression at a ratio of approximately 60:7:1 (data not shown). Thus, rather than FCR activities, these expression levels might reflect the physiological roles of VuuB, IutB and DesB.

## 6. RND Proteins in the Vulnibactin-Export System

The mechanisms of vulnibactin biosynthesis and the ferric-vulnibactin utilization system have recently been reported, but the vulnibactin-export system has not been examined yet. In *Escherichia coli*, a protein complex composed of TolC outer membrane protein, a membrane fusion protein (MFP) and resistance nodulation cell division (RND) proteins has been implicated in export of a newly synthesized enterobactin siderophore across the outer membrane [33]. A BLAST search showed two putative TolC homologs (TolCV1, VV1_0612; TolCV2, VV2_1007) and 11 putative RND proteins in the genome of *V*. *vulnificus* CMCP6. An examination of growth of deletion mutants under low-iron conditions indicated that the vulnibactin-export system comprises TolCV1 and several RND proteins, including VV1_1681 (Figure 3b) [34]. TolCV1 is absolutely necessary for the vulnibactin-export system, but the other RND proteins involved in this system remain to be investigated [34]. In multidrug-resistant pathogens, there are three categories of multidrug exporters: transporters driven by ATP hydrolysis (ABC type), drug/ion antiporters [major facilitator superfamily (MFS), multidrug and toxic compound extrusion (MATE) and small multidrug resistance (SMR) types] and tripartite transporters (RND type). AcrAB-TolC is a major RND-type multidrug exporter in Gram-negative bacteria and the crystal structure of each component of the complex has been determined [35]. Inhibitors of bacterial multidrug exporters could be potential therapeutic agents against multidrug-resistant pathogens. Furthermore, TolCV1 has been reported to be critical in RtxA1 secretion, bile salt resistance and mice lethality of *V*. *vulnificus* [36]. The vulnibactin-export system is composed of TolCV1 and several RND proteins, including VV1_1681 and the siderophore-export system might also be a target for antimicrobial agents.

## 7. Heme-Acquisition System

Heme is also a potential iron source in humans, but siderophores cannot remove iron from heme, hemin, or hemoproteins due to the iron binding affinity [37]. However, *V*. *vulnificus* can directly incorporate iron from free heme via its heme receptors [12]. We identified the heme-acquisition system in *V*. *vulnificus* M2799, with growth experiments of deletion mutants indicating that HupA and HvtA are major and minor heme receptors, respectively (Figure 3c) [38]. While VuuA is the sole receptor in the ferric-vulnibactin utilization system, HupA and HvtA function as heme receptors. Heme in the periplasm is captured by a PBP (HupB) and transported through the inner membrane by an ABC transporter (HupCD) (Figure 3c) [38]. Unlike FatB in the ferric-vulnibactin utilization system, HupB plays an essential role in the heme-acquisition system. *hupR* is located upstream of *hupA* in the opposite orientation and HupR is classified as a LysR-type positive regulator. qRT-PCR analysis using Δ*hupR* indicated that *hupA* expression, but not that of *hvtA*, is positively regulated by HupR [38]. Further studies showed that *hupA*, *hvtA* and *hupBCD* are negatively regulated by Fur (Figure 3c) [38]. Heme-utilization systems of human pathogens may be targets for antimicrobial agent and we propose HupB as a new target because of its high similarity with PBPs of other pathogens. Alice et al. first reported that *V*. *vulnificus* CMCP6 had three copies of the genes coding for TonB system and the importance of TonB12 systems in ferric-siderophore transport and the pathogenesis [39]. In many *Vibrio* species including *V*. *vulnificus* M2799, the *tonB1*-*exbB1*-*exbD1* genes were part of *hupBCD* genes, suggesting that the system plays a key role in the pathogenesis [38]. The amino acid sequence of HupA from *V*. *vulnificus* M2799 was showed 30% and 90% identities to the corresponding proteins from *V*. *vulnificus* CMCP6 (L1) and ATCC27562^T^ (L2), respectively [38]. Therefore, strain M2799 might be classified as L2 into *V*. *vulnificus* phylogenetic lineages [16]. The genome analysis of strain M2799 and the relationship between the phylogenetic lineages and the pathogenesis remains to be clarified.

## 8. New Targets for Antimicrobial Agents

Most pathogenic bacteria secrete siderophores, which are broadly classified as catecholate and hydroxamate types. In *Vibrio* iron transport, ferric-catecholate and ferric-hydroxamate siderophores are thought to bind to specific PBPs and to be reduced by specific FCRs [40]. Our results showed that VatD and IutB participate in the ferric-vulnibactin utilization system in the absence of FatB and VuuB, respectively; that is, a PBP and a FCR for a ferric-hydroxamate siderophore can also function for a ferric-catecholate siderophore. These results indicate that ferric-siderophore PBPs and FCRs may be candidates as targets for drug discovery in infectious diseases. Inhibition of ferric-siderophore PBPs or FCRs may contribute to growth retardation of most host-infectious bacteria. Structural analyses of FatB, VatD, VuuB and IutB in the ferric-vulnibactin utilization system are currently underway.

## 9. Conclusion and Future Work

To acquire iron from the environment, *V*. *vulnificus* M2799 utilizes ferric-vulnibactin, ferric-aerobactin, ferrioxamine B, heme and free ferrous and ferric ions. The ferric-vulnibactin utilization system requires ICS for vulnibactin biosynthesis and the VuuA outer membrane receptor for ferric-vulnibactin import for growth under low-iron conditions. VatD and IutB participate in the absence of FatB and VuuB, respectively. TolCV1 and several RND proteins, including VV1_1681, have functions in the vulnibactin-export system. In heme acquisition, HupA and HvtA are major and minor heme receptors and heme in the periplasm is captured by a sole PBP HupB, unlike FatB in the ferric-vulnibactin utilization system. We propose that ferric-siderophore PBPs and FCRs are candidates as targets for drug discovery in infectious diseases. Similarly to *V*. *cholerae*, *V*. *vulnificus* M2799 has the gene encoding the Feo system for transport of ferrous ions and the Fbp ABC transporter for ferric ions [41]. Genetic approaches to identify these systems are currently, underway.

## Figures and Tables

**Figure 1 marinedrugs-19-00710-f001:**
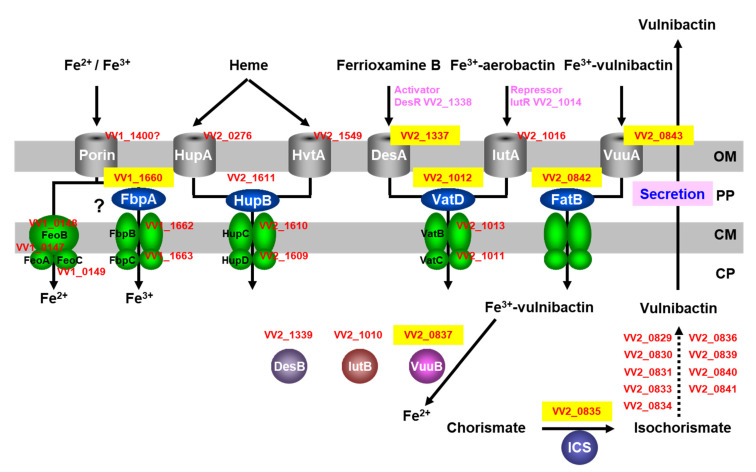
Schematic representation of the predicted iron-utilization systems in *Vibrio vulnificus* M2799. Proteins identified in PMF analysis are shown in yellow boxes. *V*. *vulnificus* CMCP6 protein numbers are shown in red letters. OP, outer membrane; PP, periplasmic space; CM, cytoplasmic membrane; CP, cytoplasm.

**Figure 2 marinedrugs-19-00710-f002:**
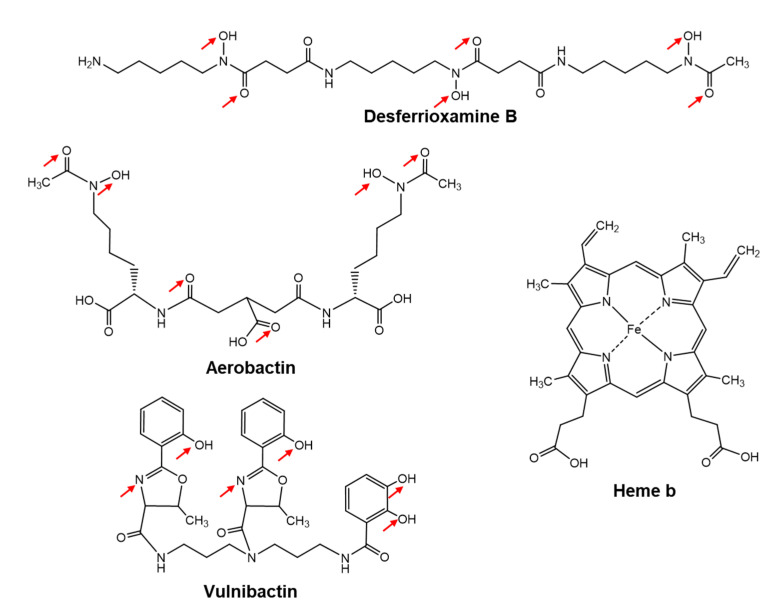
Specific iron chelators utilized by *Vibrio vulnificus* M2799. The metal ligands in the structures of the various siderophores are shown by the red arrows.

**Figure 3 marinedrugs-19-00710-f003:**
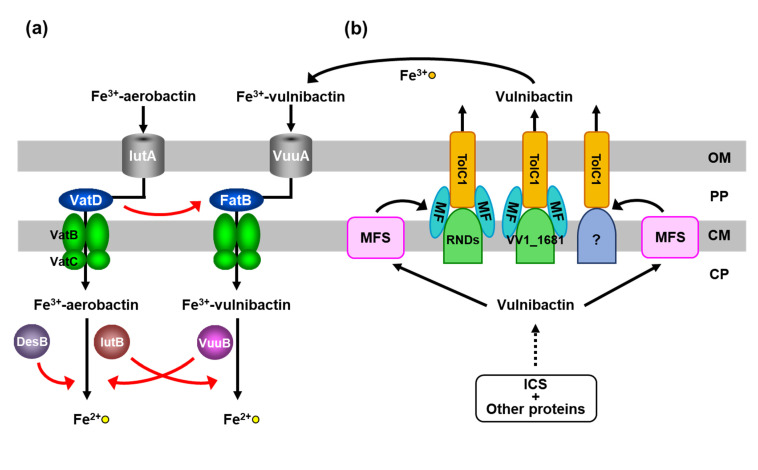
Schematic representation of vulnibactin-dependent and heme-dependent iron-utilization systems in *Vibrio vulnificus* M2799. (**a**) VatD and IutB in the ferric-vulnibactin utilization system. (**b**) RND proteins in the vulnibactin-export system. (**c**) Vulnibactin-dependent and heme-dependent iron-uptake systems are negatively regulated by Fur. OP, outer membrane; PP, periplasmic space; CM, cytoplasmic membrane; CP, cytoplasm.

**Figure 4 marinedrugs-19-00710-f004:**
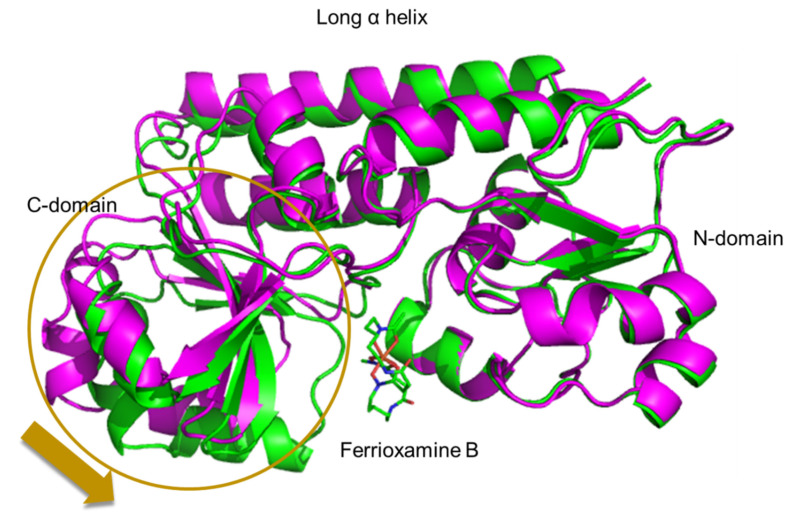
Superimposition of ferrioxamine B complex (green) and apo (magenta) forms of VatD. Ferrioxamine B is depicted as a stick model.

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
