# Peer review of "Iron-Utilization System in Vibrio vulnificus M2799"

_marinedrugs, 2021, doi:10.3390/md19120710_

Round 1
Reviewer 1 Report
This small review by Miyamoto et al about iron-utilization system is interesting but focus only on the Vibrio vulnificus M2799 clinical isolate, its iron metabolism, including transport, reduction and specific siderophores.
It is not very clear from the review what is specific or not about the clinical isolate of Vibrio vulnificus compare to other Vibrio vulnificus strains. In particular concerning the Vibriobactin biosynthesis and amount of production and secretion.
My major comment is that an introduction paragraph should be added about Vibrio vulnificus in general with some more citetation of general papers or reviews (J Oliver review for example) and then focus on M2799 clinical isolate.
Is there anything known about the IscR regulator described in this M2799 isolate and its role in iron starvation conditions to control the expression of the genes involved in hemolysin production as recently described Vibrio vulnificus (Choi et al JBC2020)
Minor comments
Line 106-108 : unnecessary details found in ref 20
Line 117 More recent references should be added about Fur and Fur in Vibrio vulnificus (Pajuerlo, Environmental Microbiology (2016) 18(11), 4005,
Fur, a transcriptional repressor that responds to iron concentrations, represses ex-116 pression of genes involved in iron-utilization systems under iron-replete conditions [25]. 117 In the Δfur mutant, the expression levels of ics, vuuB, fatB, vuuA, and vatD under both iron-118 replete and low-iron conditions were similar to those in the wild-type strain under low-119 iron conditions.
In figure 2, arrows showing the metal ligands in the structure of the various siderophore would help the reader to understand the mode of chelation
In figure 3, to be correct, Fur should be presented as dimer with one iron per subunit, not an intersubunit iron ion
Define the specificity of Vibrio vulnificus CMCP6.
The Figure 4 is not necessary because the structure can be perfectly seen in the overlapping Figure5
Liner 138-143 add the Protein database IDs for the two structures of VatD described.
Author Response
Thank you very much for your comments to our manuscript entitled “Iron-utilization system in Vibrio vulnificus M2799” (marinedrugs-1483410).
Major comments
Comment 1
It is not very clear from the review what is specific or not about the clinical isolate of Vibrio vulnificus compare to other Vibrio vulnificus strains. In particular concerning the Vibriobactin biosynthesis and amount of production and secretion. My major comment is that an introduction paragraph should be added about Vibrio vulnificus in general with some more citation of general papers or reviews (J Oliver review for example) and then focus on M2799 clinical isolate.
Response 1
According to your comment, we have added the sentences “Roig et al. had divided V. vulnificus into five phylogenetic lineages (L) according the analysis of the core genome [16]. In clinical isolates, V. vulnificus CMCP6, MO6-24, and YJ016 are classified as L1, while ATCC27562T is a member of L2.” in Introduction, and “The amino acid sequence of HupA from V. vulnificus M2799 was showed 30% and 90% identities to the corresponding proteins from V. vulnificus CMCP6 (L1) and ATCC27562T (L2), respectively [38]. Therefore, strain M2799 might be classified as L2 into V. vulnificus phylogenetic lineages [16]. The genome analysis of strain M2799, and the relationship between the phylogenetic lineages and the pathogenesis remains to be clarified.” in Section 7.
Comment 2
Is there anything known about the IscR regulator described in this M2799 isolate and its role in iron starvation conditions to control the expression of the genes involved in hemolysin production as recently described Vibrio vulnificus (Choi et al JBC2020)
Response 2
According to your comment, we have added the sentence “Recently, IscR has been reported to positively regulate vvhBA, encoding a cytolysin/hemolysin of V. vulnificus, by sensing nitrosative stress and iron starvation [14].” in Introduction. In V. vulnificus M2799, IscR has not identified. We would like to know whether IscR affects nitrosative stress and iron starvation in V. vulnificus M2799.
Minor comments
Comment 1
Line 106-108 : unnecessary details found in ref 20.
Response 1
We have deleted the sentence “V. vulnificus M2799 mutants were produced by filter mating using the suicide plasmid pDM4 [24], which was a kind gift from Prof. Debra L. Milton (Department of Molecular Biology, Umea University, Umea, Sweden).”.
Comment 2
Line 117 More recent references should be added about Fur and Fur in Vibrio vulnificus (Pajuerlo, Environmental Microbiology (2016) 18(11), 4005, Fur, a transcriptional repressor that responds to iron concentrations, represses expression of genes involved in iron-utilization systems under iron-replete conditions [25]. In the Δfur mutant, the expression levels of ics, vuuB, fatB, vuuA, and vatD under both iron-replete and low-iron conditions were similar to those in the wild-type strain under low-iron conditions.
Response 2
We have added the reference (Pajuelo, D., et al., Environ. Microbiol. 2016, 18, 4005-4012)
Comment 3
In figure 2, arrows showing the metal ligands in the structure of the various siderophore would help the reader to understand the mode of chelation.
Response 3
We have added the arrows showing the metal ligands in figure 2.
Comment 4
In figure 3, to be correct, Fur should be presented as dimer with one iron per subunit, not an intersubunit iron ion.
Response 4
We have revised Fur protein.
Comment 5
The Figure 4 is not necessary because the structure can be perfectly seen in the overlapping Figure5.
Response 5
We have deleted figure 4.
Comment 6
Liner 138-143 add the Protein database IDs for the two structures of VatD described.
Response 6
We have added PDB ID (7W8F) of VatD-ferrioxamine B complex, and the sentence “We also have determined the structure of the apo-VatD at 2.6 Å resolution, and the structural analysis at a higher resolution is in progress.” in Section 4.

Reviewer 2 Report
Comment on the manuscript “Iron-utilization system in Vibrio vulnificus M2799” (Authors: Katsushiro Miyamoto, Hiroaki Kawano , Naoko Okai , Takeshi Hiromoto , Nao Miyano , Koji Tomoo , Takahiro Tsuchiya , Jun Komano , Tomotaka Tanabe , Tatsuya Funahashi , Hiroshi Tsujibo)
An in depth, rather specific review on siderophore and heme uptake of Vibrio vulnificus M2799 is given. The focus on M2799 allowed to give an elaborate overview on the published work. However, the disadvantage is that a rather narrow view is given. What are the differences between M2799 and other highly pathogenic V. vulnificus strains like CMCP6? For instance, the TonB dependence of the outer membrane receptors is not discussed. Possibly nothing has been published on TonBs from M2799. I expect that there are close homologues in other highly pathogenic V. vulnificus strains which may be expected to behave in the same way.
In addition, it would be nice to learn a bit more about the reasons for the “100 fold higher lethality” ((Line 51). Belongs V. vulnificus M2799 to the zoonotic group and do the authors know something about its possible host?
Minor Comments
Line 57 and elsewhere: - It is not obvious why “ICS” and not the usual spelling “Ics” is used for the bacterial gene product of ics.
Line 90/91:- “also” -are these siderophores synthesized by V. vulnificus? As far as I know this is not the case.
Line 185 -187: rephrase; In some bacterial FCRs FAD is reduced with NAD(P)H.
Is there a reason why the work on DesR was not mentioned? (Biol. Pharm. Bull. 44, 1790–1795 (2021))
Author Response
Thank you very much for your comments to our manuscript entitled “Iron-utilization system in Vibrio vulnificus M2799” (marinedrugs-1483410).
Major Comments
Comment 1
What are the differences between M2799 and other highly pathogenic V. vulnificus strains like CMCP6?
Response 1
According to your comment, we have added the sentences “Roig et al. had divided V. vulnificus into five phylogenetic lineages (L) according the analysis of the core genome [16]. In clinical isolates, V. vulnificus CMCP6, MO6-24, and YJ016 are classified as L1, while ATCC27562T is a member of L2.” in Introduction, and “The amino acid sequence of HupA from V. vulnificus M2799 was showed 30% and 90% identities to the corresponding proteins from V. vulnificus CMCP6 (L1) and ATCC27562T (L2), respectively [38]. Therefore, strain M2799 might be classified as L2 into V. vulnificus phylogenetic lineages [16]. The genome analysis of strain M2799, and the relationship between the phylogenetic lineages and the pathogenesis remains to be clarified.” in Section 7.
Comment 2
For instance, the TonB dependence of the outer membrane receptors is not discussed. Possibly nothing has been published on TonBs from M2799. I expect that there are close homologues in other highly pathogenic V. vulnificus strains which may be expected to behave in the same way.
Response 2
We have added the sentences “Alice et al. first reported that V. vulnificus CMCP6 had three copies of the genes coding for TonB system, and the importance of TonB12 systems in ferric-siderophore transport and the pathogenesis [39]. In many Vibrio species including V. vulnificus M2799, the tonB1-exbB1-exbD1 genes were part of hupBCD genes, suggesting that the system plays a key role in the pathogenesis [38].” in Section 7.
Comment 3
In addition, it would be nice to learn a bit more about the reasons for the “100 fold higher lethality” ((Line 51). Belongs V. vulnificus M2799 to the zoonotic group and do the authors know something about its possible host?
Response 3
We have added the sentences “V. vulnificus M2799 showed much greater cytotoxicity than strain JCM3731 towards various cultured cells [17]. In mice inoculated with strain M2799 or JCM3731, the number of neutrophils increased, whereas strain M2799 reduced the number of macrophages, and strain JCM3731 had no effect. However, the pathogenesis of the strain is not completely elucidated.” in Introduction.
Minor Comments
Comment 1
Line 57 and elsewhere: - It is not obvious why “ICS” and not the usual spelling “Ics” is used for the bacterial gene product of ics.
Response 1
ICS is the usual spelling for isochorismate synthase (Chen Z., et al., Plant Signal Behav. 2009, 4, 493-496)
Comment 2
Line 90/91:- “also” -are these siderophores synthesized by V. vulnificus? As far as I know this is not the case.
Response 2
We have deleted the word “also”.
Comment 3
Line 185 -187: rephrase; In some bacterial FCRs FAD is reduced with NAD(P)H.
Response 3
We have rewritten the sentence “In some bacterial FCRs, reduction of oxidized FAD (FADox) to reduced FAD (FADred) was coupled with reduction by NAD(P)H.” in Section 5.
Comment 4
Is there a reason why the work on DesR was not mentioned? (Biol. Pharm. Bull. 44, 1790–1795 (2021)).
Response 4
We have added the reference (Tanabe, T., et al., Biol. Pharm. Bull. 2021, 44, 1790-1795).

Round 2
Reviewer 1 Report
The authors have taken into account the comments of my review.
The article has been improved.